# Device-Measured Desk-Based Occupational Sitting Patterns and Stress (Hair Cortisol and Perceived Stress)

**DOI:** 10.3390/ijerph16111906

**Published:** 2019-05-30

**Authors:** Gemma C. Ryde, Gillian Dreczkowski, Iain Gallagher, Ross Chesham, Trish Gorely

**Affiliations:** 1Faculty of Health Sciences and Sport, University of Stirling, Stirling FK9 4LA, UK; g.m.dreczkowski@stir.ac.uk (G.D.); i.j.gallagher@stir.ac.uk (I.G.); r.a.chesham@stir.ac.uk (R.C.); 2Department of Nursing and Midwifery, University of the Highlands and Islands, Inverness IV3 5SQ, UK; trish.gorely@uhi.ac.uk

**Keywords:** sitting, sedentary, stress, poor mental health, workplace, hair cortisol

## Abstract

*Background:* Stress and poor mental health are significant issues in the workplace and are a major cause of absenteeism and reduced productivity. Understanding what might contribute towards employee stress is important for managing mental health in this setting. Physical activity has been shown to be beneficial to stress but less research has addressed the potential negative impact of sedentary behaviour such as sitting. Therefore, the aim of this study was to assess the relationship between device-measured occupational desk-based sitting patterns and stress (hair cortisol levels (HCL), as a marker of chronic stress and self-reported perceived stress (PS)). *Methods:* Employees were recruited from four workplaces located in Central Scotland with large numbers of desk-based occupations. Seventy-seven participants provided desk-based sitting pattern data (desk-based sitting time/day and desk-based sit-to-stand transitions/day), a hair sample and self-reported perceived stress. HCL were measured using enzyme-linked immunosorbent assay and PS using the Cohen Self-Perceived Stress Scale. Linear regression models were used to test associations between desk-based sitting time/day, desk-based sit-to-stand transitions/day, HCL and PS. *Results:* There were no associations between any of the desk-based sitting measures and either HCL or PS. *Conclusions:* Desk-based sitting patterns in the workplace may not be related to stress when using HCL as a biomarker of chronic stress or PS. The relationship between sitting patterns and stress therefore requires further investigation.

## 1. Introduction

Stress can be defined as the psychological and physical state that results when an individual does not perceive they have the resources to cope with the demands and pressure of a given situation [1]. High stress levels are related to poor mental health, which is a significant contributor towards disability and mortality globally [2,3]. Within the workplace context, in addition to significant human cost, poor mental health is the leading cause of sickness absence, presenteeism and staff turnover costing UK employers alone up to £42 billion each year [4]. Understanding potential factors that are related to employee stress is therefore essential to help prevent and manage poor mental health in the workplace.

One factor that can have a positive effect on stress is physical activity. Physical activity and exercise have been shown to have a beneficial relationship with stress, with those who are more physically active reporting less subjective stress [5,6,7,8,9]. Whilst the exact reasons for this positive relationship are largely unknown, both psycho-social and biological mechanisms have been proposed [10,11]. On the other end of the activity spectrum is sedentary behaviour. Sedentary behaviour is defined as any waking behaviour characterised by a low energy expenditure ≤ 1.5 METs while sitting, reclining or lying posture [12]. Previous studies have indicated that there may be a relationship between sedentary behaviour and poor mental health [13,14,15,16]. For example, cross-sectional data from 42,469 participants of the World Health Organisation’s Study on Global Ageing and Adult Health suggests that people with depression are at an increased risk engaging in high levels of sedentary behaviour [13]. Given the beneficial relationship between physical activity and stress, and a potential relationship between sedentary behaviour and poor mental health, exploring whether sedentary behaviour has a detrimental effect on stress might be of interest.

Currently, limited research exist that has examined the relationship between sedentary behaviour and stress with what has been reported suggesting inconclusive or mixed findings. For example, one prospective cohort study looking at perceived stress and self-reported television viewing in women found that higher levels of stress were associated with watching 14–22 h of television per week but not with watching greater than 22 h [17]. Another cross-sectional study of Australian adults suggests no relationship between self-reported television viewing and self-reported perceived stress [18]. Reasons for these inconsistent findings could include the influence of contextual factors (sitting for leisure such as watching TV could actually be used as a strategy to de-stress) or the self-reported nature of the measures of stress and sedentary behaviour. In two more recent studies that have used either objective measures of stress (saliva cortisol levels or hair cortisol levels—a biomarker of chronic stress) or sedentary behaviour (body worn activity trackers), the results suggest that there is no association between stress and sedentary behaviour [19,20]. Diaz et al. (2018) measured objective sedentary behaviour using a Fitbit and self-reported end of day stress and found no association between the two outcomes [19]. Similarly, Teychene et al. (2018) found no relationship between objectively measured stress from hair cortisol levels and self-reported sedentary behaviours (television viewing, computer use, or overall sitting time) in woman from low socio-economic status areas [20].

To date, no one has examined the relationship between stress and sedentary behaviour (sitting time) using device measured sitting and objective measures of chronic stress, such as hair cortisol levels. In addition, these relationships have not been assessed in the specific context of the modern, desk-orientated workplace. The office environment is now a setting where significant amounts of sitting time is accumulated with office-based employees spending approximately 10 h sitting and not moving during the working day [21,22]. Combined with the aforementioned burden of mental health in this setting, the workplace may be an ideal context in which to investigate this issue. Therefore, the aim of this study was to assess the relationship between device-measured occupational desk-based sitting patterns (desk-based sitting time/day and number of times employees got up from their desks - desk-based sit-to-stand transitions/day) and stress (hair cortisol levels and self-reported perceived stress). It was hypothesised that greater time spent sitting at the desk and fewer sit-to-stand transitions would be associated with higher levels of hair cortisol and self-reported perceived stress.

## 2. Materials and Methods

### 2.1. Study Design and Participants

Employees were recruited from four workplaces located in Central Scotland with large numbers of sedentary and desk-based occupations and invited to take part in this cross-sectional study. Eligible participants were aged 18 years and over, worked at least 22 h per week and had been in their job or a similar role for a minimum of three months. The study consisted of a health assessment in the workplace to record physiological measures and hair samples, seven-day collection of device-measured occupational desk-based sitting time and physical activity, daily worktime log and an online survey. Data were collected from October 2016 to March 2017. Study protocols were approved by the NHS, Invasive or Clinical Research Committee (NICR16/17—paper number 4) of the University of Stirling.

### 2.2. Hair Cortisol

Hair strands roughly half the diameter of a pencil were taken from the posterior vertex of the participants head. The hair was tied with a thread and cut with professional hair dressing scissors as close to the root of the hair as possible. Samples were placed on a small piece of cardboard and wrapped in foil with a mark indicating the root end. Samples were stored at room temperature in a non-airtight container until analysis. 

Previously established methods formed the basis of the protocol for the current study [23,24]. Each sample was measured with a ruler and cut to 1 cm from the root to represent approximately the previous month of hair growth with the rest of the sample discarded. Samples were then washed for 3 min with 2.5 mL of isopropanol in a 15 mL centrifuge tube on a tube rotator and then dried overnight in a clean laminar flow hood. Hair samples were pulverised to a powder using a Retsch MM200 ball mill (12 mm grinding ball and 10 mL jar) at a frequency of 25 Hz for 1.5 min. Ninety percent of samples were prepared for single extraction and 10% of samples for duplicate extraction. The weight of each powdered hair sample was recorded using Fisher Brand MH-14 4 decimal place balance scales (to a maximum sample weight of 25 mg) and transferred into a 2 mL cryovial. To clean the hair, pure methanol (1.5 mL) was added to each sample tube and slowly rotated in the tubes in an overhead rotator for 24 h. Samples were then centrifuged in a micro centrifuge at 10,000 rpm for 2 min and 600 μL of clear supernatant transferred into a new 2 mL cryovial and methanol removed using a centrifugal evaporator. Once complete, the sample was stored in a refrigerator until analysis. Salivary cortisol buffer of pH 8 (200 μL) was then added and the samples vortexed in vials before cortisol determination was carried out using a commercially available immunoassay, Stratech High Sensitivity Salivary Cortisol EIA kit [25].

### 2.3. Occupational, Desk-Based Sitting Variables

Desk-based sitting time/day and sit-to-stand transitions/day (the number of times an employee got up from sitting at their desk) were measured using a Sitting Pad. The Sitting Pad is a research device that has been described and validated elsewhere [26]. In brief, the Sitting Pad consists of two parts; a cushion containing a pressure sensor that is placed on the office chair and a micro controller that records a time stamp to the second for each sitting and standing event. Sitting Pads were attached to employee’s chairs by GR and collected the following week (approximately seven days on the chair). Data were downloaded using a customised Sitting Pad software package (RF Technologies, Murarrie, Queensland, Australia) and exported into a spreadsheet (Microsoft Excel 2013, Microsoft). Diary data were used to exclude days when the chair may have been used by a work colleague. Work time of ≥6 h per day was classified as a valid work day, with ≥3 valid work days required for analyses to represent the majority of the working week.

### 2.4. Physical Activity Variables

Actigraph GT3X+ accelerometers were used to assess moderate to vigorous physical activity (MVPA), light activity and sedentary behaviour. Accelerometers were initialised at 30 Hz and distributed at the health assessment. Participants were advised during a one-to-one demonstration by a researcher to wear the device during all waking hours over a continuous seven-day period, on their right hip and to remove the device when in water or during contact sports. Devices were collected by GR from the workplace on the same day as the Sitting Pads. Data were downloaded using ActiLife software (version 6.13.3, Full Edition, ActiGraph, Pensacola, FL, USA) and saved as 60-second epochs. Non-wear time was removed in ActiLife using the Choi, et al. (2011) criterion [27]. Data were included if accelerometer weartime was at least 10 h per day on three days (a combination of work and non-work days to represent typical activity patterns across the week). The ActiLife software was also used to identify time spent in MVPA (≥2020 counts), light physical activity (100–2019 counts) and sedentary behaviour (0–99 counts) using establish cut points and based on one axis [28,29]. Individuals average daily time (minutes) spent in MVPA taking into account total weartime was calculated and use in statistical models. Percentage of weartime spent in MVPA, light activity and sedentary behaviour was used for descriptive data.

### 2.5. Demographic Characteristics, Anthropometric Measurements and Self-Reported Perceived Stress

Employees were sent a link via email to an online demographic survey (Jisc online surveys) seven days after the health assessment had been completed. Items in the electronic survey were used to record, sex, age, ethnic background, qualification, self-reported perceived health, employment status and annual income. Self-perceived perceived stress was measured using the 10 item Cohen Self-Perceived Stress Scale [30,31]. Self-perceived stress was derived by reversing the scores on the four positive items and then summing across all 10 items. The maximum possible score is 40 with scores greater than 20 as classed as high stress [31]. Anthropometric measurements (height and weight) were taken by trained researchers, using standardised protocols. Height was measured with a portable Marsden Leicester Height Measure stadiometer and weight was measured using Seca Sensa 804 digital scales, both with shoes removed. BMI was calculated using standard formula with >25 kg/m^2^ being defined as overweight.

### 2.6. Data Analysis

All data were analysed using SPSS version 25 (IBM, Armonk, NY, USA). Of the 131 who initially agreed to take part in the study, 110 participants provided hair cortisol data. Four of these were excluded because their values were below those that could be read by the ELISA plate (one was biologically implausible, and three were extreme outlier values for average cortisol level (>3 SD above the mean)). Crude and adjusted linear regression models were used to test associations between occupational desk-based sitting/day and desk-based sit-to-stand transitions/day, hair cortisol levels and self-reported perceived stress. Models were adjusted for self-reported level of education (University education vs. no university education), sex, daily time (minutes) spent in MVPA, Sitting Pad average day length and hours worked in the last seven days. Education and sex were included as these variables have been used in adjusted models in previous studies in this area [20] and whilst additional variables could have been included in the adjusted models (age, ethnicity, etc.), due to the small sample size, were not included to avoid overfitting the model [32].

Tests of the assumptions for linear regression analyses revealed that the data violated the assumptions of homoscedasticity and normality. Consequently, a robust regression with bootstrapping (1000 times) was employed to generate bias-corrected and accelerated 95% confidence intervals and significance tests of the model parameters. Missing data on dependent variables meant complete data was available for 77 and 76 participants for the desk-based sitting/day time and desk-based sit-to-stand transitions/day, respectively. Continuous data are presented as mean ± standard deviation. For variables that are used in statistical models (both desk-based variables and daily time spent in MVPA) the minimum and maximum values are also presented to demonstrate variability in the sample. Categorical data are presented as *n* and valid per cent.

## 3. Results

The sample is described in Table 1. Participants were mostly female, white British, full-time employed, had a university qualification, earnt between £20,001 and £50,000 per annum and perceived themselves to be in good to excellent health. The mean age was 41.8 years and mean BMI was 27.7 kg/m^2^. Participants spent most of their accelerometer total weartime in sedentary time (68%) and sat at their desks for on average 5 h 24 min per work day.

Participants in the analysed sample providing complete data were more likely to be female (chi^2^(1) = 4.4, *p* < 0.05), employed full-time (chi^2^(1) = 5.7, *p* < 0.05) and paid £20,001–£30,000 rather than >£50,001 (chi^2^(4) = 12.9, *p* < 0.05), than those who provided incomplete data. There were no other significant differences between participants providing complete and incomplete data (all *p* > 0.05). Average hair cortisol levels in the analysed sample were 19.8 ± 26.5 ng/g, range 1.7–163.56 ng/g, which was within the range previously reported for healthy adults [33,34].

The results of the analysis to test the associations between desk-based sitting/day and desk-based sit-to-stand transitions/day, and stress are shown in Table 2 (hair cortisol levels) and Table 3 (self-reported perceived stress). There were no associations between any desk-based sitting variable and hair cortisol levels or self-reported perceived stress in either crude or adjusted models. The association between self-reported stress and hair cortisol were also analysed (results not presented) and no associations were found.

## 4. Discussion

The aim of this study was to assess the relationship between device-measured occupational desk-based sitting patterns (desk-based sitting time/day and desk-based sit-to-stand transitions/day) and stress (hair cortisol, as a marker of chronic stress and self-reported perceived stress). It was hypothesised that more time spent sitting at a desk and fewer sit-to-stand transitions would be associated with higher levels of stress measured by hair cortisol levels and self-reported perceived stress. The findings of this study suggest that desk-based sitting patterns in the workplace may not be related to stress when using hair cortisol as a biomarker of chronic stress or self-reported perceived stress.

Whilst this is thought to be the first study to explore the relationship between hair cortisol levels and device-based measures of occupational sitting patterns, previous studies employing self-reported measures of these outcomes have reported similar findings [19,20]. For example, although they used self-reported sedentary behaviour in different context (not specifically occupational sitting) and reported on a different population (*n* = 72 women from low socio-economic areas of Australia as opposed to a more educated working population in Scotland), Teychenne et al. (2018) found no relationship between self-reported TV viewing time, computer usage or overall sitting and hair cortisol levels [20]. They suggested that the use of a self-report measure of sedentary time could have contributed towards this finding. However, in the current study, a device was used to assess occupational sitting patterns and still no relationship with hair cortisol level or perceived stress was found. Whilst further studies are required to confirm the findings reported in the current study, other explanations for the lack of findings, not related to the measure of sitting time, may be needed.

One such explanation could be related to the measures of stress—hair cortisol and self-reported perceived stress. It might be that hair cortisol levels as a measure of stress might be too general to answer the question proposed by the current research as it does not take into account the complexity of the relationship between the exposure to stress and the stress response. Hair cortisol levels provide a general or “global” measure of chronic stress with each centimetre of growth roughly equivalent to one month of hair cortisol build up. It therefore represents an individual’s accumulated response to stress regardless of the source i.e., pressures at work, pressures at home and other life events all combined. The same is true for the perceived stress scale which again is a non-specific, global measure of stress. Knowing more details about the source of stress might therefore be important. This idea is alluded to in a recent study by Diaz et al. (2018) [19]. In this N of one, yearlong observational study (a type of statistical modelling that determines associations at an individual level and accounts for individual variability) [35], they assessed device recorded sedentary behaviour, a global measure of stress (self-reported end of day stress) and source specific stress (including running late for work) in 79 healthy working adults in America. Whilst they did not measure hair cortisol levels, Diaz et al. reported no association between device measured sedentary time and self-reported end of day stress but a negative association between device measured sedentary time and running late for work. They also reported significant inter-individual variability between sedentary behaviour and both the levels and source of stress. Their finding and the results from the current paper reinforce the complexity of the stress relationship and suggest that what people are doing whilst sedentary might be more important for stress than being sedentary itself.

This could also be the case for why there is a relationship shown in the video gaming literature between sitting whilst playing a game and stress. Sedentary behaviour research has drawn on this body of literature when assessing the relationship between stress and sedentary time [20]. Some of this research has shown that sitting playing video games is associated with increased hair cortisol levels and it was presumed this could be as a result of the sitting behaviour [36]. However, it could be suggested that “playing the video game” itself which includes other factors such as competition effects [37] might be more likely to be responsible for the relationship reported with stress than the sitting itself. In the context of the workplace, it could be the task that the employee is doing on the computer whist sitting and not the sitting itself that modifies the relationship with stress. Completing the same tasks whilst sitting compared to using a standing desk in an experimental design might help uncover the relationship between these factors.

The current study provides an interesting basis from which to explore this area further. The strengths of this study are the use of a device measure of sitting patterns and assessing the specific context of the desk. Whilst using hair cortisol levels could also be seen as a strength, as previously discussed, this may also have been a limitation in that it cannot determine the source of the stress. Future studies should look at repeating this work in larger samples, different populations and using a wider range of stress measures including source specific stress, in addition to hair cortisol levels. Another limitation was that occupational desk-based sitting patterns were measured over seven days and the 1 cm hair sample would represent one month of growth. Future studies should adopt a more time congruent design and methods such as assessing sitting or sedentary time over a longer time period and accounting for additional factors than can influence hair cortisol levels such as seasonal variations, hair characteristics, medical conditions and use of specific drugs [33,38].

## 5. Conclusions

This study provides preliminary evidence to suggest that desk-based sitting patterns in the workplace (desk-based sitting time/day and sit-to-stand transitions/day) may not be related to stress (hair cortisol levels, as a marker of chronic stress and self-reported perceived stress). These findings suggest that global measures of stress such as hair cortisol levels might be too general a measure to assess the relationships between sitting time and stress. They also suggest that the source of stress and what the individual is doing whilst sitting might be a more important consideration than the sitting itself. This relationship between sitting patterns and stress therefore requires further investigation.

## Figures and Tables

**Table 1 ijerph-16-01906-t001:** Description of the entire eligible sample and analysed sample with complete desk-based sitting pattern data.

Characteristics	Entire Eligible Sample (*n* = 131)	Complete Desk-Based Sitting Data (*n* = 77)
**Age** *(years) (n = 117; n = 76)*	41.8 ± 9.8	40.8 ± 9.7
**Sex ***		
Female	93 (71)	60 (78)
Male	38 (29)	17 (22)
**BMI** *(kg/m^2^) (n = 126; n = 80)*	27.7 ± 6.0	27.9 ± 5.9
**Ethnic background**		
White British	109 (98.2)	75 (98.7)
Other	2 (1.8)	1 (1.3)
**Qualification**		
University or higher	72 (63.7)	52 (67.5)
No university qualification	41 (36.3)	25 (32.5)
**Employment status ***		
Full time	101 (88.6)	72 (93.5)
Part time	13 (11.4)	5 (6.5)
**Hours worked in the last 7 days** *(hours) (n = 113; n = 77)*	40.2 ± 9.8	40.7 ± 8.7
**Perceived health**		
Good to excellent	110 (96.5)	75 (97.4)
Poor to fair	4 (3.5)	2 (2.6)
**Annual Income ***		
<£20,000	10 (8.8)	6 (7.8)
£20,001–£30,000	35 (31.0)	29 (37.7)
£30,001–£40,000	21 (18.6)	16 (20.8)
£40,001–£50,000	19 (16.8)	14 (18.2)
>£50,001	28 (24.8)	12 (15.6)
**Desk-based variables** *(n = 104; n = 77)*		
Desk based sitting *(hours/day)*	5.4 ± 1.3 (2.2–8.8)	5.3 ± 1.3 (2.2–8.8)
Desk based sit-to-stand transitions *(number/day)*	21.2 ± 10.5 (6.0–66.3)	21.1 ± 9.5 (6.0–61.5)
**Physical activity and sedentary variables***(n = 125; n = 77)* (% weartime)		
Sedentary behaviour	68.2 ± 6.9	68.1 ± 6.1
Light physical activity	27.8 ± 6.3	28.2 ± 5.9
MVPA	4.0 ± 2.6	3.7 ± 2.0
*(Mins/day)*		
Daily time spent in MVPA	33.67 ± 23.28 (5.5–183.5)	31.85 ± 17.42 (5.9–84.4)

Categorical data are presented as *n* (valid %). Continuous data are presented at mean ± standard deviation. Exact n for each variable is presented next to the variable names. For variables that are used in statistical models (both desk-based variables and daily time spent in MVPA) the minimum and maximum values are presented in brackets after the mean ± standard deviation. * Significant differences between samples at *p* < 0.05.

**Table 2 ijerph-16-01906-t002:** Associations between desk-based sitting, desk-based sit-to-stand transitions and hair cortisol levels.

Desk Based Sitting Variable (mins/day)	Crude Models				Adjusted Models ^a^			
	b	SE B	β	*p* Value	b	SE B	β	*p* Value
**Desk based sitting**	0.02 (−0.05, 0.08)	0.04	0.07	0.467	0.04 (−0.06, 0.13)	0.05	0.11	0.435
**Desk based sit-to-stand transitions**	−0.18 (−0.78, 0.57)	0.37	−0.06	0.661	−0.28 (−1.1, 0.77)	0.44	−0.09	0.618

^a^ Models adjusted for sex, qualification, daily time (minutes) spent in MVPA, Sitting Pad average day length and hours worked in the last seven days. B—unstandardized beta, SE B—standard error for the unstandardized beta, β—standardized beta.

**Table 3 ijerph-16-01906-t003:** Associations between desk-based sitting, desk-based sit-to-stand transitions and perceived stress levels (from those who also had hair cortisol).

Desk Based Sitting Variable (mins/day)	Crude Models				Adjusted Models ^a^			
	b	SE B	β	*p* Value	b	SE B	β	*p* Value
**Desk based sitting**	−0.01 (−0.03, 0.01)	0.01	−0.10	0.404	−0.01 (−0.03, 0.01)	0.01	−0.15	0.249
**Desk based sit-to-stand transitions**	0.02 (−0.13, 0.21)	0.09	0.03	0.770	0.01 (−0.16, 0.22)	0.11	0.01	0.945

^a^ Models adjusted for sex, qualification, daily time (minutes) spent in MVPA, Sitting Pad average day length and hours worked in the last seven days. B—unstandardized beta, SE B—standard error for the unstandardized beta, β—standardized beta.

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
