# Peer review of "Device-Measured Desk-Based Occupational Sitting Patterns and Stress (Hair Cortisol and Perceived Stress)"

_ijerph, 2019, doi:10.3390/ijerph16111906_

Round 1
Reviewer 1 Report
This paper explores an interesting and under-studied question regarding the association between workplace sitting and stress. the objective measures of sitting and stress response add methodological rigor to this study.
Several concerns must be addressed.
- Please check the grammar and some word choices throughout the paper. For example, "ill psychological health" might be a better choice that "mental ill-health". Also, on line 135, please write out the word "percent" since it is the first word in the sentence. You should not begin a sentence with a symbol.
- The use of hair cortisol to examine chronic HPA axis is interesting and the duration of the study to explore "average" sitting behaviors is also an interesting combination of methods. These methods seem well thought out and well executed.
- On line 37, this point may be very important. The key MAY be in previous authors subjective way of measuring stress and the impact of intentional physical activity. Research appears to show that those who cope with stress using physical activity respond less negatively to the stress. But, I do not follow how sitting more in the workday will be associated with more stress. You need to provide a more robust rationale for conducting this study. Does the literature support it? If not, state very clearly that it has not been explored.
- On line 45, please consider your use of the word "positive" in the context of a scientific communication. The word "positive" connotes a positive correlation which would not be the case between stress and physical activity. Please think about using the word "beneficial" or another alternative.
- Beginning on line 73, this sentence is not stating your hypothesis of the underlying mechanisms in an effective manner. You need to rework this sentence to more effectively describe why you believed there was an association between sitting and stress at work. This is difficult but you should look to the literature to see if anyone has studied the association between sitting at work and autonomy or workload.
- Statistical methods are appropriate.
Overall this paper needs a bit of attention and some editing. However, despite the negative finding it explores an important topic using objective measures. This reviewer feels it takes the next logical step from other similar studies and does contribute to the existing literature.
Author Response
Response to Reviewer Comments
Thank you for your comments. Please find below our response to your points and indications of where manuscript adaptations have occurred. Our responses are coloured in blue.
Reviewer 1
This paper explores an interesting and under-studied question regarding the association between workplace sitting and stress. The objective measures of sitting and stress response add methodological rigor to this study.
Several concerns must be addressed.
Please check the grammar and some word choices throughout the paper. For example, "ill psychological health" might be a better choice that "mental ill-health".
The language we chose was to reflect that of the previous work referenced in the introduction. We have changed it to a more standard term which is that of ‘poor mental health’ which we hope will reflect the reviewers comment and still be true to the literature we reference.
Also, on line 135, please write out the word "percent" since it is the first word in the sentence. You should not begin a sentence with a symbol.
Agreed. This has been changed.
The use of hair cortisol to examine chronic HPA axis is interesting and the duration of the study to explore "average" sitting behaviors is also an interesting combination of methods. These methods seem well thought out and well executed.
Thank you.
On line 37, this point may be very important. The key MAY be in previous authors subjective way of measuring stress and the impact of intentional physical activity. Research appears to show that those who cope with stress using physical activity respond less negatively to the stress. But, I do not follow how sitting more in the workday will be associated with more stress. You need to provide a more robust rationale for conducting this study. Does the literature support it? If not, state very clearly that it has not been explored.
This section has been updated to more clearly illustrate what the previous literature has said and a stronger rationale presented. We have drawn more on the fact that it was the positive relationship reported with PA that influenced our logic in assessing sitting and stress. This now reads as follows:
One factor that can have a positive effect on stress is physical activity. Physical activity and exercise have been shown to have a beneficial relationship with stress, with those who are more physically active reporting less subjective stress [5-9]. Whilst the exact reasons for this positive relationship are largely unknown, both psycho-social and biological mechanisms have been proposed [10, 11]. On the other end of the activity spectrum is sedentary behaviour. Sedentary behaviour is defined as any waking behaviour characterized by a low energy expenditure ≤ 1.5 METs while sitting, reclining or lying posture [12]. Previous studies have indicated that there may be a relationship between sedentary behaviour and poor mental health [13-16]. For example, cross-sectional data from 42,469 participants of the World Health Organization's Study on Global Ageing and Adult Health suggests that people with depression are at an increased risk engaging in high levels of sedentary behaviour [13]. Given the beneficial relationship between physical activity and stress, and a potential relationship between sedentary behaviour and poor mental health, exploring whether sedentary behaviour has a detrimental effect on stress might be of interest.
On line 45, please consider your use of the word "positive" in the context of a scientific communication. The word "positive" connotes a positive correlation which would not be the case between stress and physical activity. Please think about using the word "beneficial" or another alternative.
Agreed. This has been changed to beneficial.
Beginning on line 73, this sentence is not stating your hypothesis of the underlying mechanisms in an effective manner. You need to rework this sentence to more effectively describe why you believed there was an association between sitting and stress at work. This is difficult but you should look to the literature to see if anyone has studied the association between sitting at work and autonomy or workload.
The main aim of this paper was to assess whether there was a relationship between sitting and stress. We hypothesis that high sitting would mean high stress. There were many valid reasons why we thought we should explore this relationship which included the fact that there was the opposite relationship reported for PA. However, even in PA, it is still not clear why this is the case and we have added a sentence in the introduction stating this. As for sitting, whilst we proposed a ‘mechanism’ for our hypothesis (workload and autonomy), and on reflection, this was potentially premature. Whilst we could add this to the discussion where it would be better suited, given the lack of relationship reported, this does not seem appropriate. This point has therefore been removed from the introduction.
Statistical methods are appropriate.
Many thanks.
Overall this paper needs a bit of attention and some editing. However, despite the negative finding it explores an important topic using objective measures. This reviewer feels it takes the next logical step from other similar studies and does contribute to the existing literature.
Many thanks.
Reviewer 2 Report
Thank you for the opportunity to review this important manuscript on the possible relationship between occupational sitting and physical and psychological manifestations of stress.
When using accelerometry to discern sedentary time from light activity PA bouts and light intensity from moderate intensity PA bouts, cut points need to be carefully chosen. On line 133, the authors claim to have chosen both cut points (sedentary/light and light/moderate) based on reference 27. However, that reference only gives the cut point for light/moderate, not a cup point for sedentary/light. I would encourage the authors to explain how this cut point was chosen. Further, as this cut point is very low, it is possible that the participants were even more sedentary than it would appear with the cut point set so low. This raises the question whether these participants varied sufficiently in PA level to test the study hypothesis. A quick regression analysis to examine whether PA predicted stress may help elucidate this.
This question of the adequacy of representation of putative predictors is also raised by the low percentage moderate to vigorous activity (3.6% among those who completed the study. This again raises the question if there were participants in the study who had PA levels sufficient to impact their perceived or biological stress levels. Furthermore, it is entirely unclear from the data presented, how much variability there was between participants in PA levels. Since both outcome measures are at the person level, prediction would only be feasible if a wide range of person-related predictors were represented among the participants. Based on the limited data presented, however, it would appear that participants were fairly homogeneous on predictor variables. It is, however, statistically impossible to use such near constants to predict variable outcomes.
A similar problem exists with the main predictor variables, desk based sitting and sit-to-stand transitions. It appears participants were quite similar especially on these variables. A better test of the study hypothesis would be obtained, had the study contrasted, e.g. office workers who use active motion sitting devices or treadmill desks with those who do not. In other words, occupational sedentariness may not have been sufficiently disrupted in this study to affect stress but that may be because the vast majority of participants had disruptions in occupational sedentariness below the threshold needed to affect their stress levels.
In the discussion, the authors may want to add a consideration of occupational sitting variables as mediators rather than sufficient predictors of physical and psychological stress manifestations. In other words, given, say the task to make a stressful phone call, physical and psychological manifestations of stress may be moderated by an employee's PA during the call (say if pedaling on a seated active workstation or walking on a treadmill vs. sitting still). Many consider sitting still not to be stressful by itself (and indeed many do so during their leisure hours to 'relax', even if PA is a better active stress buster (i.e. moderates physical and psychological manifestations of stress given exposure to stressful demands).
With additional caveats and added to the manuscript taking into account the above, the manuscript makes an important contribution to the literature on occupational sedentariness.
Author Response
Response to Reviewer Comments
Thank you for your comments. Please find below our response to your points and indications of where manuscript adaptations have occurred. Our responses are coloured in blue.
Reviewer 2
When using accelerometry to discern sedentary time from light activity PA bouts and light intensity from moderate intensity PA bouts, cut points need to be carefully chosen. On line 133, the authors claim to have chosen both cut points (sedentary/light and light/moderate) based on reference 27. However, that reference only gives the cut point for light/moderate, not a cup point for sedentary/light. I would encourage the authors to explain how this cut point was chosen.
The cut points were based on the Troiano 2008 reference which used 0-99 for sedentary but didn’t make it explicit in the paper. The original paper for this reference was Freedson 1998. This has now been added so it is clear where the cut point is from (Ref 28 and 29).
Further, as this cut point is very low, it is possible that the participants were even more sedentary than it would appear with the cut point set so low.
The 99 count sedentary cut point is widely used. The other commonly used cut point is 150 which we agree would have allocated more of the time to sedentary. However, there is no consensus on which is the best. With the sedentary time variable not used in the analysis and only as a descriptor there would be rationale for using either cut point and this would not change MVPA which was used in our models and not sedentary behaviour.
This raises the question whether these participants varied sufficiently in PA level to test the study hypothesis. A quick regression analysis to examine whether PA predicted stress may help elucidate this. This question of the adequacy of representation of putative predictors is also raised by the low percentage moderate to vigorous activity (3.6% among those who completed the study. This again raises the question if there were participants in the study who had PA levels sufficient to impact their perceived or biological stress levels. Furthermore, it is entirely unclear from the data presented, how much variability there was between participants in PA levels. Since both outcome measures are at the person level, prediction would only be feasible if a wide range of person-related predictors were represented among the participants. Based on the limited data presented, however, it would appear that participants were fairly homogeneous on predictor variables. It is, however, statistically impossible to use such near constants to predict variable outcomes. A similar problem exists with the main predictor variables, desk based sitting and sit-to-stand transitions. It appears participants were quite similar especially on these variables.
The reviewer has raised several very valid points with regards to variability of the data which we will aim to address here. We understand the need for variably in the sample to be able to show an effect. The concern was that the sample, with regards to outcomes measures and predictors, were not varied enough. Firstly, the most important variables to show variance in is that of the outcome measures from the sitting pad. We report the mean and standard deviation of these variables and both of the desk-based sitting variables show variation. For example, the standard deviation on the sitting data was over an hour per day and the transitions were around 10 per day and very similar to our previous study using this device which we published a paper on highlighting the significant variability in this measure:
Ryde GC, Brown HE, Gilson ND, Brown WJ. Are we chained to our desks? Describing desk-based sitting using a novel measure of occupational sitting. Journal of Physical Activity and Health. 2014; 11(7):1318-1323
In addition, the desk-based sitting time in the analysed sample had a minimum of 2.2 hours/day and a maximum of 8.8 hrs per day again highlighting the variation in the current sample.
As for the physical activity data, we again believe this to be variable. The minimum and maximum for daily time spent in MVPA in the analyzed sample was 5.9 minutes per day and 1 hour 24 minutes per day respectively. Again, the variance can be seen in the mean and standard deviation. To make this clear to the reader the outcome variables used in the statistical models are now shown in Table 1 with the min and max values included.
A better test of the study hypothesis would be obtained, had the study contrasted, e.g. office workers who use active motion sitting devices or treadmill desks with those who do not. In other words, occupational sedentariness may not have been sufficiently disrupted in this study to affect stress but that may be because the vast majority of participants had disruptions in occupational sedentariness below the threshold needed to affect their stress levels.
There are several ways this hypothesis could have been tested and we agree a stronger design would have been experimental. We have suggested this as a potential future study in the discussion: Line 253 - Comparing the same tasks whilst sitting and using a standing desk in an intervention design might help uncover the relationship between these factors.
In the discussion, the authors may want to add a consideration of occupational sitting variables as mediators rather than sufficient predictors of physical and psychological stress manifestations. In other words, given, say the task to make a stressful phone call, physical and psychological manifestations of stress may be moderated by an employee's PA during the call (say if pedaling on a seated active workstation or walking on a treadmill vs. sitting still). Many consider sitting still not to be stressful by itself (and indeed many do so during their leisure hours to 'relax', even if PA is a better active stress buster (i.e. moderates physical and psychological manifestations of stress given exposure to stressful demands).
The reviewer raises and interesting point for future research to try and establish whether work tasks or the sitting itself is. This links back to the first reviewers comments on the mechanisms behind the relationship between sitting and stress. Given no relationship was found in the current research, the points already included in the discussion regarding improvements in study design to assess the question of interest is there a relationship, are more relevant for this current paper.
With additional caveats and added to the manuscript taking into account the above, the manuscript makes an important contribution to the literature on occupational sedentariness.
Many thanks.
Round 2
Reviewer 1 Report
Reviewer concerns have been adequately addressed.